# Novel Optical Kerr Switching Photonic Device Based on Nonlinear Carbon Material

**DOI:** 10.3390/mi14122216

**Published:** 2023-12-08

**Authors:** Ke Wang, Zhoufa Xie, Jianhua Ji, Yufeng Song, Bin Zhang, Zhenhong Wang

**Affiliations:** College of Electronics and Information Engineering, Shenzhen University, Shenzhen 518060, China; wong@szu.edu.cn (K.W.); 2210434002@email.szu.edu.cn (Z.X.); jjh@szu.edu.cn (J.J.); yfsong@szu.edu.cn (Y.S.); bin_zhang13@u.nus.edu (B.Z.)

**Keywords:** 2D material, all-optical signal processing, Kerr optical switching

## Abstract

In the context of current communication systems, there is an urgent demand for more efficient and higher-speed optical signal processing technologies. Researchers are actively exploring new materials and devices to harness nonlinear optical phenomena, seeking advancements in this field. Nonlinear carbon materials, especially promising 2D materials, have garnered attention for their potential interaction with light and have become integral to the development of all-optical signal processing devices. This study focuses on utilizing a photonic device based on a nonlinear Au/CB composite material for optical Kerr switching. The application of Au/CB as a nonlinear material in the Kerr switch represents a noteworthy advancement, demonstrating its capability to modulate optical signals. By appropriately applying a pump light, the study achieves optical Kerr switching with an extinction ratio of approximately 15 dB in the fully off state of the signal light carrying a 10 GHz analog signal, marking a pioneering achievement in the field to the best of our knowledge. The experimental results, encompassing extinction ratios, signal control, and stability, not only validate the feasibility of this technology but also underscore its potential applicability within optical communication systems. The successful modulation and control of a 10 GHz analog signal showcase the practicality and effectiveness of the Au/CB-based optical Kerr switch. This progress contributes to the continuous evolution of optical Kerr switching, a crucial component in modern optical communication systems. Therefore, we believe that the Au/CB-based optical Kerr switch is an exceptionally promising and stable all-optical signal processing device. As the contemporary communication landscape evolves, the integration of this technology holds the potential to enhance the efficiency and speed of optical signal processing.

## 1. Introduction

Fiber optics communication has become a crucial part of modern communication systems. However, the conversion and transmission of optical signals still heavily rely on electronic devices. This constraint limits the switching rates between electronic and optical signals, which has led researchers to explore all-optical signal processing techniques to overcome these limitations [1,2]. One of the most promising approaches in this field is the investigation of nonlinear optical phenomena. Nonlinear optical effects arise when a material’s response to incident optical signals exhibits nonlinearity relative to signal intensity [3]. This presents a potential pathway for all-optical signal processing through the exploitation of nonlinear effects. Nonlinear material properties have already been utilized to create optical switches, frequency converters, signal modulators, and other essential devices in optical communication systems. However, further research is necessary to enhance the performance of these optical devices due to limitations such as material stability, nonlinear coefficients, and cost considerations. The advancement of these optical devices is crucial, as it will lead to advanced functionalities in optical communication systems.

The seminal discovery of graphene [4] in 2004 marked the advent of a new era in materials science, particularly in the realm of two-dimensional materials. Graphene has attracted widespread attention due to its outstanding electrical, mechanical, and optical properties. Consequently, researchers have initiated investigations into a range of novel 2D materials with similar characteristics, including but not limited to graphene [5,6], graphyne [7,8], black phosphorus (BP) [9,10], transition metal dichalcogenides [11], perovskite calcium titanate, MXenes [12,13], and MBenes [14]. These two-dimensional materials have garnered significant interest due to their unique nonlinear optical properties, positioning them as promising candidates for advanced photonic devices. In practical terms, the ability to precisely control and manipulate light signals with unprecedented accuracy is of paramount importance. All-optical switches facilitate the routing of optical signals, optical modulators enable signal modulation for data encoding, and ultrafast pulse lasers find extensive applications in high-speed data transmission and medical diagnostics, among others [15,16,17,18,19,20,21,22,23,24,25]. These functionalities play a pivotal role in the design and optimization of optical communication systems, with the choice of materials for these devices exerting a profound impact on their overall performance.

In an all-optical switch, the switching of light between two or more output channels or the modulation of light within a single channel is controlled by an optical signal, a role performed by an electric field in electro-optic switches. In electro-optic switch devices, the change in refractive index is induced by an applied electric field, while in all-optical switch devices, the refractive index change is generated by variations in the optical field intensity, termed the pump signal [26]. This study primarily focuses on the generation of optical switches in compact devices by combining two-dimensional materials with strong nonlinear Kerr effects with optical fibers.

In 2012, Maryam S. Sakhdari [26] and colleagues utilized carbon nanotube composite materials as a nonlinear layer in metamaterial structures. Through simulation calculations based on effective optical parameters, they achieved a low-threshold all-optical switch within the visible light range using the composite material. In 2015, Chen et al. successfully developed a device known as the Topological Insulator-Clad Microarray (TCM) by depositing Bi_2_Te_3_ nanosheets on a microarray [27]. This device exhibited exceptional performance in Kerr switch applications, achieving a remarkable extinction ratio of up to 14 dB with a control power of merely 250 mW and a working range extending beyond 20 nm. This research marked a significant milestone in the field of optical switches. However, despite the promise shown by the Bi_2_Te_3_ material, it also revealed certain limitations. In 2017, Zheng and Yang’s research team introduced black phosphorus (BP) into the realm of all-optical signal processing, creatively achieving an impressive Kerr switch extinction ratio of up to 26 dB [28]. Nevertheless, BP material possesses a fundamental drawback, its susceptibility to oxidation, which results in rapid degradation of its electronic and optical properties, severely limiting its long-term prospects. While substantial progress has been made, the quest for more stable materials remains a critical challenge.

In 2018, researchers such as Song and Chen for the first time prepared a few-layer antimony (Sb) material and applied it to all-optical Kerr switches. Utilizing a micro-substrate covered with this material, they successfully achieved an extinction ratio of approximately 12 dB [29]. This indicated the potential of antimony in all-optical switching, although further research is required to enhance its performance. In 2019, Wang and Zheng, among others, fabricated an optical Kerr switch based on bismuth-based materials, achieving an extinction ratio of approximately 22 dB with a control power of 320 mW [30]. This demonstrated the promising application prospects of bismuth-based materials in the field of optical switches. However, these materials also come with their limitations, necessitating further research to address these issues. Similarly, in 2019, researchers such as Wang and Chen introduced black phosphorus quantum dots into all-optical Kerr switches, achieving an extinction ratio of approximately 20 dB [31]. This indicated the potential of black phosphorus quantum dots in optical switching, and their relative stability compared with traditional black phosphorus materials is noteworthy. However, it was not until 2023 that Liu and Wang’s research team achieved a significant breakthrough by utilizing a TiN/Ti_3_C_2_ heterostructure device to attain a 27 dB extinction ratio at a control power of 200 mW [32]. This marked a major advancement. It is important to note that most of the aforementioned studies achieved success under conditions where the signal light had relatively high transmittance, which still poses certain limitations in practical applications. Consequently, future research efforts must remain dedicated to addressing the challenges in optical signal processing to better meet the demands of real-world applications.

Recently, the size-dependent nonlinear optical properties of carbon black (CB) have been reported [33]. Considering the potent surface plasma enhancement capabilities of gold nanoparticles, researchers are exploring whether combining Au nanoparticles with carbon black could result in a more prominent nonlinear optical phenomenon. In this study, mesoporous carbon black-supported Au nanoparticles (Au/CB) were synthesized to investigate the optical characteristics of this complex. Carbon black was synthesized using a literature-reported method, followed by the loading of Au nanoparticles via a wet chemistry process. Subsequently, we explore the intricate design and characterization of a Kerr switch photonic device based on the nonlinear Au/CB composite material. This approach offers an innovative solution to address the urgent needs of modern optical communication systems. The application of a 320 mW pump light source represents a significant milestone in our research, enabling the attainment of an extinction ratio of approximately 15 dB in the fully closed state of the signal light. Simultaneously, we successfully demonstrated precise control over a 10 GHz analog signal using this photonic device. These experimental achievements underscore the immense potential of harnessing the distinctive nonlinear optical characteristics of the Au/CB composite material.

## 2. Materials and Characterization

The potential of the Au/CB composite material for nonlinear optical applications was assessed by conducting a comprehensive characterization. The material was analyzed using the scanning electron microscope (SEM), dynamic light scattering (DLS), attenuated total reflection (ATR), and near-infrared spectroscopy (NIR) methods to determine the corresponding properties. The Au loading was 2 wt % (“2 wt %” stands for “2 weight percent,” indicating that the specified substance or component constitutes 2% of the total weight of the material), and Figure 1A,B show that the incorporation of Au into CB did not change the morphology of CB. Figure 1C,D show the DLS results, revealing that CB and Au/CB share a similar size and zeta potential. Furthermore, particle size analysis revealed that Au/CB exhibited particle sizes ranging from ~28.3 to 74.8 nm, while CB displayed sizes from ~24.1 to 74.8 nm. Moreover, the Au/CB exhibited a significantly higher zeta potential, measuring at 35.1 mW, compared with CB’s 22.6 mW. Moreover, conclusions drawn from techniques such as attenuated total reflection infrared (ATR-IR) and near-infrared (NIR) absorption spectroscopy indicate that under the same distribution, the blank value of Au/CB is higher. Therefore, the light absorption of Au/CB is superior, as shown in Figure 1D,F. Thus, the decoration of CB with Au nanoparticles enhanced the absorption ability for NIR light without damaging the intrinsic structures of CB. Meanwhile, X-ray absorption spectroscopy was employed to confirm the presence of gold in the sample. As shown in Figure 2A, gold species with an average valence of +0.5 were observed, demonstrating that Au species were incorporated in the CB sample. The slightly positive charge suggests electron transfer from Au to CB and serves as a clue for stabilizing the sample. To check the size of the Au species, Figure 2B provides the coordinating information of Au, and a ~10 Au-Au coordination number was concluded based on the fitting result of Figure 2A, indicating that the average size of Au species is around 3 nm.

In summary, through the analyses of SEM, DLS, ATR, and NIR, we provide compelling evidence for the potential of the Au/CB composite material in nonlinear optical applications. The high zeta potential, small particle size, and distinct near-infrared absorption collectively indicate the characteristics of the Au/CB material. These findings hold significant promise for utilizing this material in optical devices within the specified wavelength range, contributing to the advancement of 2D materials in optical communications.

## 3. Au/CB-Based Optical Kerr Switcher

### 3.1. Theory of the Kerr Switch

The optical Kerr Switch is a device that helps to control optical signals by utilizing the optical Kerr effect. This nonlinear optical phenomenon is based on the relationship between light intensity and the refractive index of a medium during the propagation of light in nonlinear materials [34]. The Kerr switch is an essential component in an all-optical network and holds great promise for numerous applications. Figure 3 illustrates its working principle.

The Kerr all-optical switch based on the Au/CB-microfiber composite structure consists of an Au/CB-microfiber composite structure and an optical polarizer. A pump light and a signal light are simultaneously injected into the Au/CB-microfiber composite structure, with an optical polarizer connected at the output end. The purpose is to convert the polarization rotation into a change in optical power. When the pump light is off, the polarization state (SOP) of the signal light is essentially orthogonal to the polarizer, effectively suppressing the transmission of the signal light. When the pump light is absent or weak while the signal light is on, the polarization axis of the signal light is orthogonal to the polarizer, blocking the signal light. With the gradual increase in pump light intensity, the nonlinear interaction between the pump light and the signal light is enhanced, resulting in additional nonlinear phase shifts in the Kerr nonlinearity of the Au/CB-microfiber composite structure. Due to the large nonlinear refractive index of the Au/CB-microfiber composite structure and the strong interaction of mutual light, the signal light may undergo cross-phase modulation, causing significant polarization rotation of the signal light after propagating along the Au/CB-microfiber composite structure. Therefore, due to the cross-phase modulation caused by the pump signal, the polarization state of the detection signal will undergo a significant change, intentionally controlled by the intensity of the pump signal. At an appropriate pump power, when the polarization state of the signal light is parallel to the polarizer, the transmittance of the signal light reaches its maximum. The observed phenomenon bears a resemblance to polarization rotation seen in highly nonlinear optical fibers [35]. This sequential manipulation of polarization states and the ensuing nonlinear effects contribute to the modulation and control of signal light, elucidating the operational principles of the optical Kerr switch.

In the schematic diagram, when the pump light is turned on, the pump light generates birefringence effects as it passes through the Au/CB-microfiber composite structure. This results in different changes in the refractive indices of the signal light in the horizontal and vertical components. At the output end, the phase difference between the horizontal and vertical components manifests as a change in the polarization state of the signal light when it is output through the Au/CB-microfiber composite structure. A portion of the signal light passes through the optical polarizer due to the change in polarization state. The transmittance of the signal light is correlated with the optical intensity of the pump light. Therefore, by adjusting the output power of the pump light, control over the on/off state of the signal light can be achieved.

Therefore, the formula expressing the transmittance and phase relationship of the signal light can be given as follows:(1)$$T=\sin^{2}\left(\frac{∆\emptyset }{2}\right)$$

In this equation, T represents the transmittance of the signal light, and ∆∅ is the phase difference between the pump light and the signal light, with the formula for the phase difference given as follows:(2)$$∆\emptyset =\frac{2\pi L}{\lambda }\left({\tilde{n}}_{x}-{\tilde{n}}_{y}\right)$$

Due to the wavelength difference between the pump light and the signal light, where λ represents the wavelength of the signal light, the refractive indices are modified as follows: ${\tilde{n}}_{x}=n_{x}-∆n_{x}$ and ${\tilde{n}}_{y}=n_{y}-∆n_{y}$; the linear refractive indices, $n_{x}$ and $n_{y}$, are often different due to mode birefringence. The nonlinear components of the refractive indices, $∆n_{x}$ and $∆n_{y}$, also show differences due to birefringence caused by the control light. Furthermore, $∆n_{x}$ and $∆n_{y}\;$ can also be expressed as:(3)$$∆n_{x}=2n_{2}{\left|E_{p}\right|}^{2}$$
(4)$$∆n_{y}=2n_{2}b{\left|E_{p}\right|}^{2}$$
where $n_{2}\;$ represents the nonlinear refractive index, b is a parameter related to the third-order nonlinear susceptibility, and if the third-order nonlinearity is solely due to electrons, then b=1/3. $E_{p}\;$ represents the power of the pump light. Therefore, the phase difference of the signal light can be determined using the following formula:(5)$$∆\emptyset =\frac{2\pi L}{\lambda }\left[n_{x}-n_{y}-2n_{2}\left(1-b\right){\left|E_{p}\right|}^{2}\right]$$

### 3.2. Experimental Procedure and Discussion of Results

Figure 4 shows the experimental setup for Kerr switching. The external cavity laser ECL1 emits continuous wave (CW) light at 1552 nm as the pump light, while ECL2 is the signal light fixed at 1550 nm. Two polarization controllers (PCs) are used to adjust the state of polarization (SOP) for the pump and signal lights. An Erbium-doped fiber amplifier (EDFA) amplifies the pump light. To suppress amplified spontaneous emission (ASE), an optical band-pass filter (OBPF) is used with a bandwidth of about ±0.25 nm. The pump and signal lights are then coupled through a 3 dB optical coupler (OC) and co-propagate through the Au/CB-coated microfiber. Before turning on the pump light, the SOP of the signal light is aligned to be almost orthogonal relative to the polarizer, which effectively blocks the signal light to the maximum extent and achieves the maximum extinction ratio for the Kerr switcher.

The results of the experiment, which are shown in Figure 5, compare the transmission changes in signal light between Au/CB-coated and bare microfibers under two conditions: with the pump light on and off. In Figure 5a, when the pump light is off, the signal light remains open. By adjusting the polarization controller, the signal light is completely blocked, and the signal light is turned off, similar to the Kerr switch being in the off state. When the pump light is turned on and its power is increased, the polarization angle of the signal light changes. At a pump light power of 320 mW, accounting for insertion losses, the optical power entering the Au/CB-coated microfiber is reduced to 142 mW. At this point, some of the signal light passes through the polarization controller, and the spectrometer shows a signal light intensity of about −55 dBm. At this point, it corresponds to the Kerr switch being in the on state. But upon turning off the pump light again, the signal light intensity rapidly drops back to the fully blocked cutoff state before the pump light is turned on. The extinction ratio of the signal light under the conditions of pump light off and on is approximately 15 dBm (from −70 to −55 dBm). When replacing the sample with a CB composite structure while keeping other conditions unchanged, when the pump light power reaches 320 mW, the signal light intensity reaches about 60 dBm, and the extinction ratio is approximately 10 dBm. This indicates that the introduction of Au nanoparticles enhances the optical Kerr effect of Au/CB. We attribute this outcome to the potential enhancement of nonlinear optical response through the support of localized surface plasmon resonance (LSPR) by Au nanoparticles. Previous studies have highlighted that gold nanoparticles, by generating a strong local electric field through the support of LSPR, can enhance nonlinear optical responses. This enhancement effect of the local electric field contributes positively to the optical Kerr effect, mitigating to some extent the inhibitory effects induced via nonlinear absorption [36,37].

To validate the operational functionality of the Kerr switch within the Au/CB microfiber, we conducted experiments using a regular microfiber without Au/CB material as a control. The results, depicted in Figure 5b, illustrate that, regardless of the pump light status, the transmittance of the signal light remains constant. To evaluate the switching performance at varying transmittance levels (Figure 5c), we adjusted the polarization controller to achieve a signal light transmittance of −55 dBm with the pump light turned off. Subsequently, upon activating the pump light and reaching a power level of 320 mW, the signal light intensity increased from −55 dBm to −41 dBm, resulting in an extinction ratio of approximately 14 dBm. Notably, the switching effect is more pronounced in the signal light with lower transmittance. For comparative purposes, a conventional single-mode microfiber device was employed, as illustrated in Figure 5d. Irrespective of the pump light status, the signal light intensity remained constant, indicating the incapability of a regular microfiber to achieve optical Kerr switching. These experimental findings robustly support the premise that pump light passing through the Au/CB-covered microfiber induces nonlinear polarization rotation in the signal light. The polarization state of the signal light is intricately linked to the intensity of the pump light, aligning with the theoretical derivations. In the experiment, varying the pump power from 100 mW to 340 mW revealed a corresponding variation in the transmittance of the signal light, as shown in Figure 6. As the pump power increases, the signal light transmittance also increases. It is worth noting that at 117 mW, the signal light passes through and the Kerr switch light activates. When the pump power reaches 320 mW, the signal light transmittance reaches its peak intensity, measuring at around 15 dBm. To affirm the capability of the signal light carrying information to pass through by altering its transmittance, we conducted the experiment depicted in Figure 7a,b. Applying a 10 GHz RF-modulated analog signal to the signal light, both when the transmittance was fully blocked and at approximately −55 dBm, showcased spectral sideband details. It is evident that the signal light with lower transmittance carries information with better quality, affirming the practical feasibility of this device for real-world applications.

To investigate the performance of the optical Kerr switch at different wavelength intervals, we kept the pump light fixed at a wavelength of 1552 nm while tuning the signal light wavelength from 1549 nm to 1556 nm. The pump light power was maintained at 320 mW, and the signal light power was kept constant. As shown in Figure 8, the extinction ratio of the signal light remained stable at approximately 14–15 dBm with an error of about 1 dBm when the signal light wavelength was changed from 1549 nm to 1556 nm. In addition, with the signal light power held constant, the pump light power was maintained at 320 mW, the wavelength was set at 1550 nm and 1552 nm, and stability measurements were conducted on the device every 5 min within a one-hour timeframe, as illustrated in Figure 9. The extinction ratio of the signal light remained stable at 15 dBm throughout the entire duration, which demonstrated the high stability of the device. To compare with similar experiments in the past, Table 1 summarizes the performance parameters of optical Kerr switches that are based on different 2D materials. This study achieved a modulation depth of 15 dBm for the first time when the signal light was completely blocked and demonstrated the control of 10 GHz analog signals. Additionally, the Au/CB composite material can still be used normally after being stored at room temperature for two months. This strongly proves that the Au/CB-based optical Kerr switch can operate stably for a long time under high-speed conditions.

We achieved a significant breakthrough in optical Kerr switches by obtaining a 15 dB extinction ratio using a compact row photon device that combines Au/CB with microfiber. This innovative device completely blocks the signal light, unlike other nonlinear two-dimensional materials that rely on increasing the transmittance of the signal light when it reaches a certain intensity. Additionally, this device allows for precise control of 10 GHz analog signals. Carbon-based materials offer high stability, making them ideal for the integrated processing and storage of our device. This progress in optical Kerr switches highlights the practical application potential of Au/CB in optical communication devices and provides critical theoretical support for the future design of optical devices.

## 4. Conclusions

In summary, we demonstrated through comprehensive material characterization that Au/CB is an outstanding nonlinear material. By effectively harnessing the nonlinear Kerr effect of the Au/CB composite material in conjunction with microfiber technology, we successfully implemented an all-optical Kerr switch for signal processing. The experimental results indicate that, among various 2D materials, Au/CB achieved, for the first time, an extinction ratio of 15 dBm when the signal light was completely blocked, with a pump light power of 320 mW. The device successfully modulated and controlled a 10 GHz analog signal, showcasing practical switching functionality. Furthermore, Au/CB exhibits significantly higher stability compared with other 2D materials. Therefore, the Au/CB-based optical Kerr switch emerges as a stable and highly promising all-optical signal processing device. It holds the potential to become a key component in modern optical communication systems, such as wavelength-division multiplexing (WDM) systems and optical switching networks, as well as in the broader field of all-optical signal processing. This research is of significant importance in driving the development of high-performance devices to meet the growing demands of modern communication systems.

## Figures and Tables

**Figure 1 micromachines-14-02216-f001:**
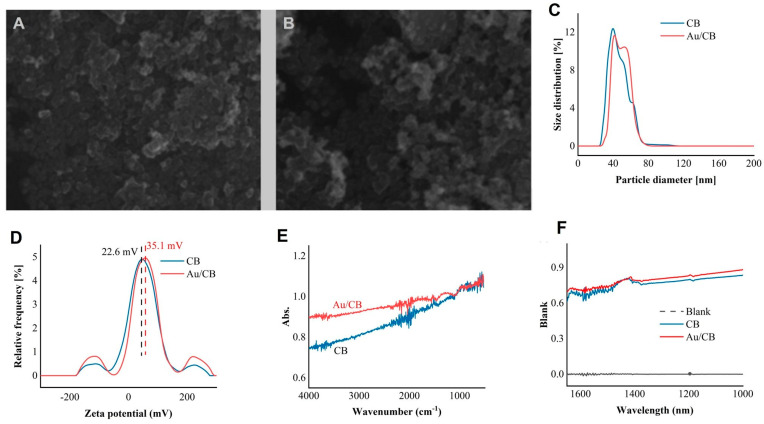
Characterization of CB and Au/CB materials: (**A**) scanning electron microscope (SEM) image of CB; (**B**) SEM image of Au/CB; (**C**) size distributions; (**D**) zeta potential values; (**E**) attenuated total reflectance infrared spectroscopy (ATR-IR) profiles; and (**F**) optical absorption behaviors of CB and Au/CB.

**Figure 2 micromachines-14-02216-f002:**
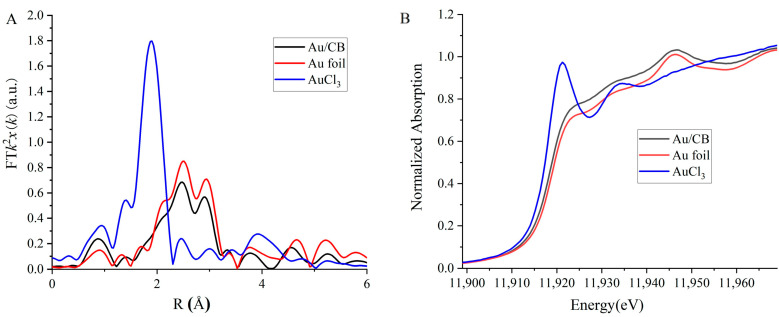
XAS analysis of Au L3 edge of Au/CB. (**A**) X-ray absorption near-edge structure (XANES) spectra; (**B**) extended X-ray absorption fine structure (EXAFS) analysis.

**Figure 3 micromachines-14-02216-f003:**
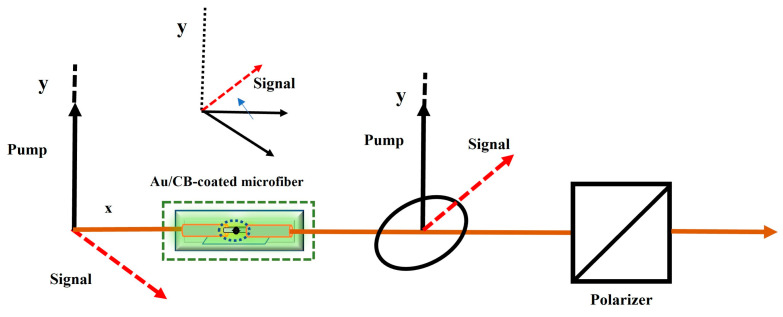
Au/CB-coated 6 μm micro-nanofiber Kerr switch schematic diagram. The red arrow represents the polarization state rotation of the signal light during the operational process.

**Figure 4 micromachines-14-02216-f004:**
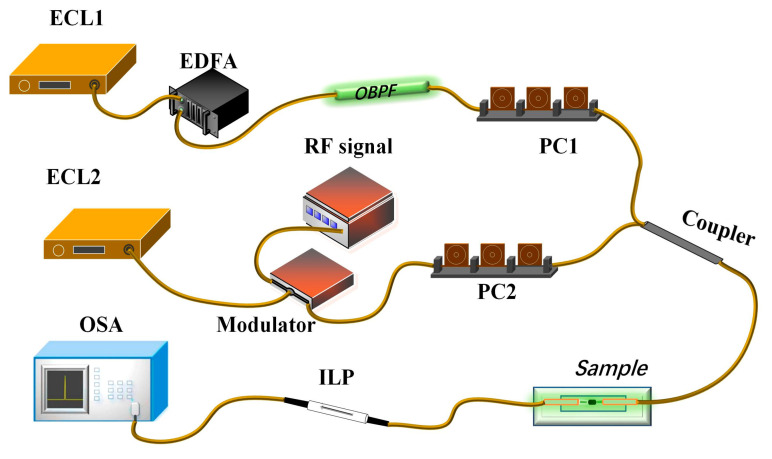
The experimental setup diagram for the optical Kerr switch.

**Figure 5 micromachines-14-02216-f005:**
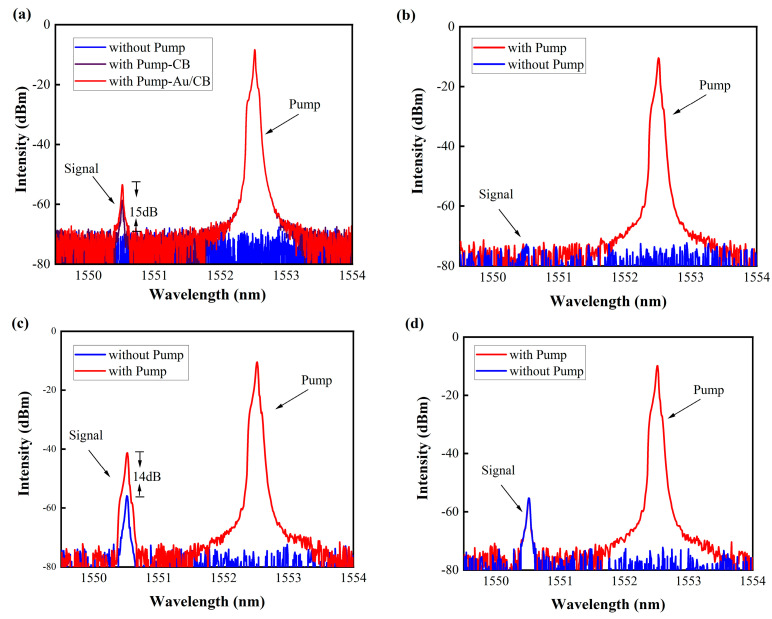
The signal light output intensity under conditions of the pump light being on and off. (**a**) Describes the output of the signal light in Au/CB and CB samples when the signal light is completely blocked; (**b**) describes the output of the signal light in bare fiber; (**c**,**d**) illustrate the output of the signal light in Au/CB and bare fiber when the signal light transmission is at −55 dBm.

**Figure 6 micromachines-14-02216-f006:**
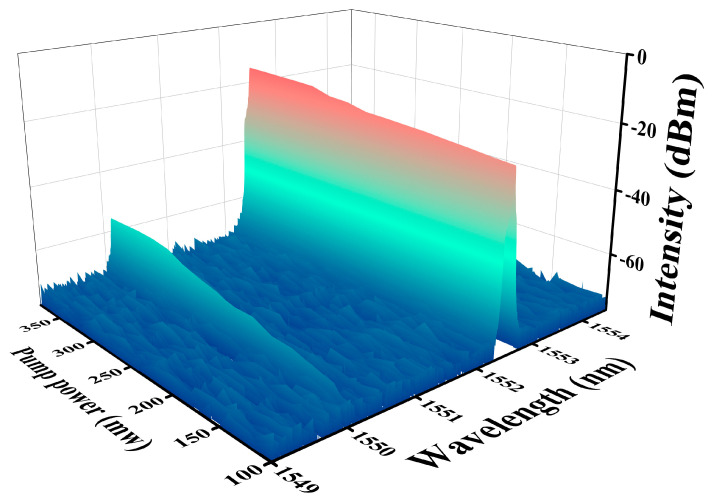
The output of the signal light slowly increases the power of the control light from 100 to 360 mW.

**Figure 7 micromachines-14-02216-f007:**
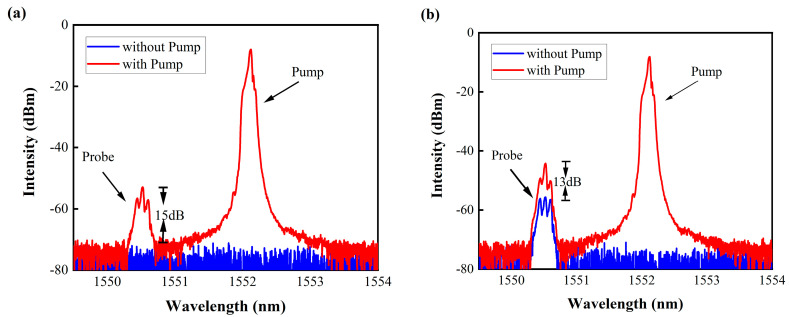
A 10 GHz RF-modulated analog signal of the signal light, where (**a**) indicates that the signal light is completely blocked, and (**b**) indicates that the signal light has a transmittance of approximately −55 dBm.

**Figure 8 micromachines-14-02216-f008:**
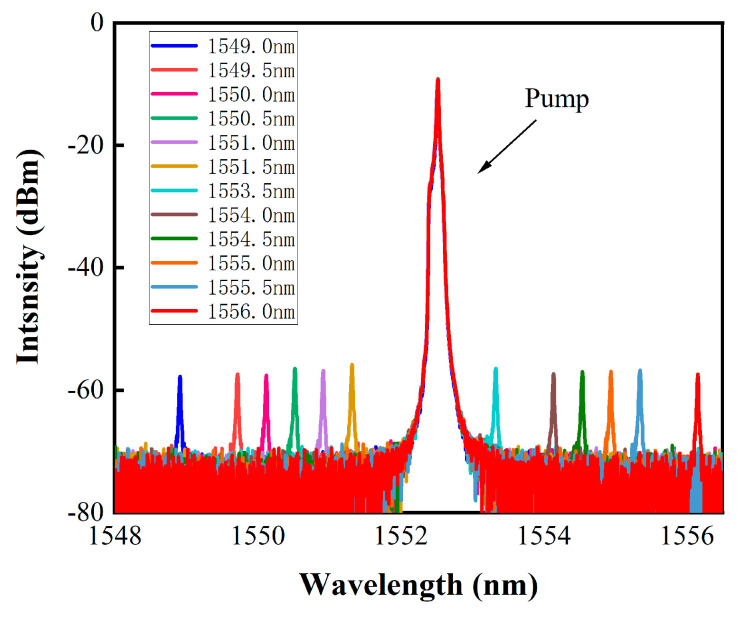
The relationship between the pump light and signal light of different wavelengths.

**Figure 9 micromachines-14-02216-f009:**
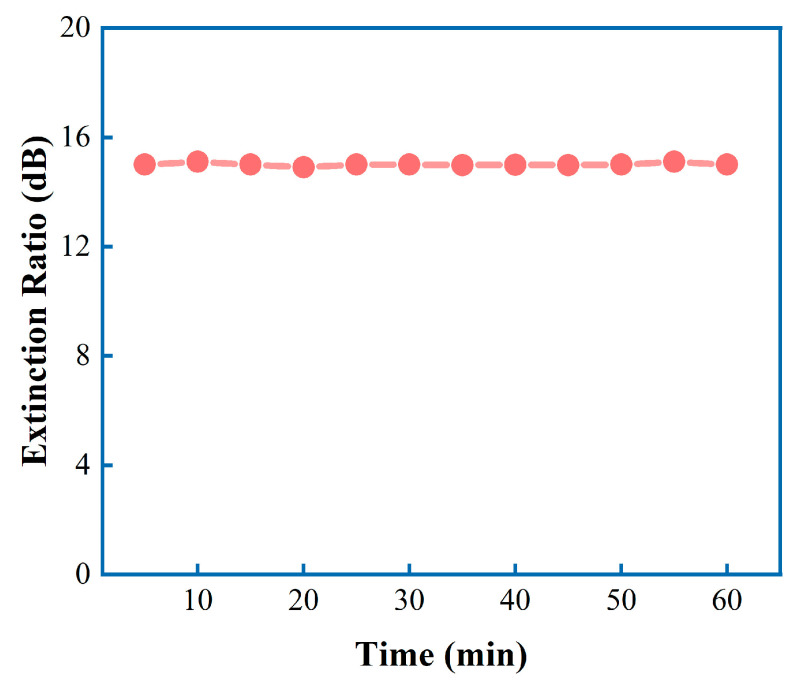
The stability error line of this device every 5 min.

**Table 1 micromachines-14-02216-t001:** Performance parameters of Kerr switches utilizing various 2D materials.

Sample	Intensity (dBm)	Modulation Depth (dB)	Stability	Signal Frequency	Pump Power (mW)
TCM [27]	-	14	-	-	250
BP [28]	−53.6~−26.9	26	<2 days	-	300
FLA [29]	−50~−37	13	-	-	316
FLB [30]	−59.6~−37.6	22	-	-	320
BPQD [31]	−56~−36	20	<2 weeks	-	160
THM [32]	−60~−33	27	>1 month	-	200
Au/CB	–~−55	15	>2 month	10 GHz	320

## Data Availability

Data are contained within the article.

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
