# Peer review of "Novel Optical Kerr Switching Photonic Device Based on Nonlinear Carbon Material"

_micromachines, 2023, doi:10.3390/mi14122216_

Round 1

Reviewer 1 Report

Comments and Suggestions for Authors

Comments on

 Novel Optical Kerr Switching Photonic Device Based on Non-linear Carbon Material  

by Ke Wang et al

The article describes a set of experimental studies of the optical Kerr switch performance. The paper is written in an easy to comprehend way, the results are timely and solid and are definitely of a great interest to the scientific community.

I have a few textual and formatting comments that should be easy to implement in the new version of the paper:

L16, 28, 31 (and many more): please, make an non-breakable space between the number and unit,  and/or between a word and citation, i.e 10GHz -> 10 GHz, etc

L40-41: “garnered widespread attention and ushered in a range…”  please, rephrase this statement, I can’t get the meaning of it

L91: carbon black (CB): CB has been already used several times

L102:  material material

Figure 1 caption:  In fact the caption is missing…. You have sub-captions for (a)-(f) plots, but not the general caption.

Figure 1: SEM is not defined

L111: 2 wt % : What is that?

L111: “showed that…”: figures 1(a) and 1(b) do not show that the incorporation of Au doesn’t change the morphology. That is your conclusion. In order to arrive to this conclusion, you should explain what is shown in the photo, what one would expect to see in case the morphology-change hypothesis holds, etc…  Please, provide this discussion.

L119: “better optical absorption for Au/CB, as shown in Figures 1(d) and 1(f)…”: I see almost the same red/blue distributions in these Figures. So, this conclusion is wrong.

L139: “signal light”: there is no signal light in the Figure 2, only the probe. Is it the same? You have the same issue in the text, sometimes you use word ‘signal’, sometimes ‘probe’. If that is not the same – define both.

L139-151: this paragraph is impossible to follow. Please, rewrite to make it clear.

L158-159:  “Due to the wavelength..”: it must be a point at the end of equation, if you start the sentence with the capital letter.

L161: n_x and n_y are not defined

L162: polarisation component of what?

Eq (3) and (4): looks like  \delta n_x=\delta n_y. Is it true?

L168: What is the dimension of E_p? Looks like it is dimensionless, which is strange…

Eq (5): there is a problem in the math. It is impossible to get to (5) from (2-4)

Figure 4: The quality of the picture is extremely poor. Please, replace. The caption is inadequate. The pictures show distributions of intensity vs wavelength, not the comparison.

L205-232: The discussion in this paragraph is impossible to follow. Please, re-write.

Figure 5,6,7: Please, place the figures in relevant parts of the text. And provide a proper caption.

References: The list of citations is rather old, that is strange for such an important topic. The only new cited work is [27] that looks like a self-citation.

In summary, I recommend this manuscript for the publication after the authors have addressed my comments.

Reviewer 2 Report

Comments and Suggestions for Authors

The manuscript ID micromachines 2732892 mainly presents a study about Kerr switch photonic system based on nanoscale Au and Carbon with enhanced nonlinearities. Please see below a list of comments to the authors.

1. It is not clear the selection of carbon black for this study.

2. How was selected the size of the Au nanoparticles?

3. Is there an influence of the shape of the Au nanoparticles in the main findings?

4. Please comment how is determined the volume fraction of the metal in the hybrid structures. The authors are invited to discuss and see other examples for instance: https://doi.org/10.3390/photonics7030054

5. Do the nonlinearities of the Au/CB are dependent on incident polarization?

6. Is there an influence of the nonlinear optical absorption in the main findings?

7. The advantages and disadvantages of the device proposed must be confronted with updated publications in the topic. You can see for instance: https://doi.org/10.1016/j.physe.2012.11.001

8. Experimental data to provide evidence of the incorporation of the Au in the samples studied are welcome.

9. Since no typical z-scan experiments or multi-wave mixing evaluations were carried out, it is requested a solid support to guarantee the presence of the Kerr effect instead of any other phenomena.

10. Please report the ablation threshold an the error bar in experimental data 

Comments on the Quality of English Language

A proofreading is mandatory

Round 2

Reviewer 2 Report

Comments and Suggestions for Authors

The solution of the authors for fundamental points raised in the review stage still requires deeper details within the text. Please see below:

*My major concern is the potential influence of nonlinear optical absorption. Besides, it is not clear the selection of carbon black for this study considering only the Kerr effect. The reference [33] proposed in the explanation includes optical limiting properties exhibited by carbon black systems. And no information about this crucial phenomenon of nonlinear absorption in switching has been analyzed in the work.

**The influence of size and shape of the nanoparticles to obtain plasmonic or protoplasmonic response dependent on incident polarization in the system is not described.

**No discussion about determination of volume fraction of the metal in the hybrid structures and support of the influence of this parameter in the main findings has been provided as suggested.

Comments on the Quality of English Language

A proofreading is suggested

Round 3

Reviewer 2 Report

Comments and Suggestions for Authors

*My major concern about the potential influence of nonlinear optical absorption to the Kerr effect is still present, and then I have to reiterate my recommendation.

The nonlinear optical absorption can inhibit the magnitude of the optical Kerr effect and the participation of different carbon-based samples must show different switching properties if absorptive nonlinearity is expected. The authors are invited to estimate how can be the influence of a nonlinear optical absorption in the switching effect taking into account comparative carbon-based systems to guarantee the potential applications proposed. You can see for instance

http://dx.doi.org/10.1016/j.physe.2015.05.035

and

https://doi.org/10.1016/j.carbon.2021.10.054

Comments on the Quality of English Language

A proofreading is suggested

Round 4

Reviewer 2 Report

Comments and Suggestions for Authors

The authors have made an effort to further describe the potential of the Kerr nonlinearities. The manuscript can be useful for future research and then I can recommend this work for publication in present form.

Comments on the Quality of English Language

A proofreading is suggested.